# G-Protein β-Subunit Gene *TaGB1-B* Enhances Drought and Salt Resistance in Wheat

**DOI:** 10.3390/ijms24087337

**Published:** 2023-04-15

**Authors:** Xin-Xin Xiong, Yang Liu, Li-Li Zhang, Xiao-Jian Li, Yue Zhao, Yan Zheng, Qian-Hui Yang, Yan Yang, Dong-Hong Min, Xiao-Hong Zhang

**Affiliations:** State Key Laboratory of Crop Stress Biology for Arid Areas, College of Agronomy, Northwest A&F University, Yangling 712100, China

**Keywords:** wheat, G-protein β-subunit, *TaGB1*, abiotic stress

## Abstract

In the hexaploid wheat genome, there are three Gα genes, three Gβ and twelve Gγ genes, but the function of Gβ in wheat has not been explored. In this study, we obtained the overexpression of *TaGB1* Arabidopsis plants through inflorescence infection, and the overexpression of wheat lines was obtained by gene bombardment. The results showed that under drought and NaCl treatment, the survival rate of Arabidopsis seedlings’ overexpression of *TaGB1-B* was higher than that of the wild type, while the survival rate of the related mutant *agb1-2* was lower than that of the wild type. The survival rate of wheat seedlings with *TaGB1-B* overexpression was higher than that of the control. In addition, under drought and salt stress, the levels of superoxide dismutase (SOD) and proline (Pro) in the wheat overexpression of *TaGB1-B* were higher than that of the control, and the concentration of malondialdehyde (MDA) was lower than that of the control. This indicates that *TaGB1-B* could improve the drought resistance and salt tolerance of Arabidopsis and wheat by scavenging active oxygen. Overall, this work provides a theoretical basis for wheat G-protein β-subunits in a further study, and new genetic resources for the cultivation of drought-tolerant and salt-tolerant wheat varieties.

## 1. Introduction

Wheat (*Triticum aestivum* L.) is affected by various abiotic stresses during its growth and development, such as drought, salinity, heat, cold, heavy metals and the ozone [1]. Among them, droughts and soil salinization are the main factors limiting the increase in wheat yield and quality [2]. When plants were subjected to abiotic stresses such as drought, saline–alkali, extreme temperature, etc., the stimulation of external stress factors recognizes, transforms and transmits these stimulation signals through a series of complex signal transduction networks, and finally induces the expression of specific functional genes, and then regulates the biochemical and physiological reactions to respond to these stimuli [3]. Studies have shown that the G-protein is a sensor involved in various important biological pathways. It plays an essential role in the response of plants to abiotic stress by participating in various signal pathways and different signal transduction levels. 

Heterotrimeric G-proteins (G-proteins) are composed of Gα (40–46 kDa), Gβ (37–44 kDa) and Gγ (6–9 kDa) subunits [4], the latter two of which are referred to as the βγ complex. In general, the Gα subunit contains a Ras-like domain and a helical domain. The Gβ subunit contains a coiled-coil domain in the N-terminal region and has seven WD40 repeats. The Gγ proteins are subdivided into three types and the A-type is defined by a CaaX motif, whereas the B-type is similar to the A-type, but it lacks the CaaX motif. C-types contain enriched cysteine residues at the C-terminal and N-terminal, similar to the canonical Gγ subunits [5,6]. As crucial signal transducers, G-proteins have attracted increasing attention in the field of signal transduction. According to the classical paradigm, G-protein signaling is initiated by the association of extracellular ligands with GPCR (or RGS) and subsequently inducing a conformational change, which promotes the exchange of GDP for GTP associated with the Gα subunit. GTP-bound Gαβγ heterotrimeric complex tends to dissociate into a Gα-GTP monomer and Gβγ heterodimer [7]. The dissociated units of Gα-GTP monomer and Gβγ heterodimer can interact with a wide array of specific downstream effectors of distinct signaling pathways [8]. 

Plant G-proteins are involved in multiple aspects of plant growth, including seed germination, seedling development, and biotic and abiotic stresses. For instance, Arabidopsis *GPA1* plays a positive role in the signal of three ion channels (K^+^, Ca^2+^, anion) that ABA inhibits in the stomatal opening [9,10], and is also a positive regulator of plant cell division [11,12]. *AGB1* is involved in the modulation of cell proliferation in hypocotyls [12,13], as well as organ differentiation [14]. It positively regulates the salt tolerance of plants [15] and the defense responses to fungal infection [16,17], and modulates apoptosis caused by tunicamycin [18]. *AGG3* is a positive regulator of stress responses [19]. Maize Gα subunits (CT2) control meristem development [20], and the Gβ subunit controls shoot meristem development and immune responses [21]. Cucumber *CsGPA1* regulates seed germination and early seedling development, including hypocotyl elongation and root growth [22]. *CsGG3.2* positively regulates the chilling tolerance of cucumber [23]. In rice, *RGA1* participates in the regulation of plant growth and development. The disruption of *RGA1* leads to the dwarfing of upright leaves [24,25,26]. *RGB1* cooperates with *RGA1* in cellular proliferation and seed fertility [27]. *RGB1* is also a positive regulator of drought adaption, whereas *qPE9-1* is modulated by *RGB1* and functions as a negative regulator in the drought stress response [28]. The overexpression of *RGG1* can enhance the salt tolerance of plants [29]. In pea, the G-protein members, PsGα and PsGβ, were expressed in large amounts under drought, high temperature and high salt treatment. The Gβ subunit has a role in the nitric-oxide-induced stomatal closure in response to heat and drought stress [30,31]. The Gγ subunits, GW2 homologues, exist in other grain crops (such as corn, wheat, and sorghum), and increase the cell number of spikelet shells to participate in regulating the seed weight [32,33,34].

At present, many related research results of G-protein have been reported in succession, among which the study of Arabidopsis G-protein is more thorough. However, in wheat, the function of wheat G-protein β-subunits has not been determined, especially since they participate in the regulation of abiotic stress response. The homology between TaGB1 and AGB1 was as high as 93.46%, indicating that TaGB1 and AGB1 may have some similarities in function [35]. Therefore, we will further analyze the functions of *TaGB1*. Arabidopsis plants overexpressing *TaGB1-B* were obtained by inflorescence infection, and wheat lines overexpressing *TaGB1-B* were obtained by gene bombardment. The results showed that the overexpression of *TaGB1-B* could enhance the tolerance of Arabidopsis and wheat to drought and salt stress. This study provides evidence for the involvement of *TaGB1-B* in the response of plants to drought and salt stress, and provides useful information for the genetic improvement of drought and salt resistance of wheat.

## 2. Results

### 2.1. Expression Profile Analysis of Wheat TaGB1 Homologs

An Ensembl Plants database search showed that *TaGB1* exists in three chromosomal groups of A, B and D. Therefore, we named the three homologs *TaGB1-A*, *-B* and *-D*. The homologies of the three homologs in the coding sequence are 99.33%, and the homologies of the amino acid sequence are 99.91%. In order to further analyze the difference of expression of these three homologs under stress treatment, we also downloaded the relative expression abundance of *TaGB1* homologues in the leaves of 7-day-old seedlings under drought-, heat- and PEG-simulated drought stress from the wheat expression database (Appendix A). Then, we analyzed the expression levels of *TaGB1* genes under different stress conditions. From the overall trend, the expression level of all three homologs is similar (Figure 1), but due to the large difference in the expression levels of *TaGB1-B* under different stress treatments, we finally decided to select *TaGB1-B* for follow-up research.

### 2.2. Expression Patterns of TaGB1-B in Different Tissues under Stress

The expression level of *TaGB1-B* in different stress and hormone treatments was detected by qRT-PCR. The results showed that the expression of *TaGB1-B* changed under stress and hormones. We found that the expression of *TaGB1-B* changed under the stress and hormone treatments, it shows a trend of rising first and then falling as a whole (Figure 2a–i). The results showed that *TaGB1-B* was involved in abiotic stress response. Then, we continued to evaluate the expression level of *TaGB1-B* in wheat seedling tissue under drought and salt treatment. The results showed that *TaGB1-B* was differentially expressed in roots, stems and leaves (Figure 2j,k). Drought treatment upregulated *TaGB1-B* in roots and leaves, and its expression was the highest after 6 h of stress, showing about 3.5 times the upregulation. Under salt stress, the expression of *TaGB1-B* in roots, stems and leaves showed little difference, and the expression showed an upward trend, reaching the peak at 12 h, and slowly returning to the initial expression level after 24 h.

### 2.3. Overexpression of TaGB1-B Enhances Drought and Salt Resistance of Arabidopsis

To study the effects of *TaGB1-B* on plants under drought or salt stress, we selected a homozygous T_3_ generation, *TaGB1-B,* overexpressing the Arabidopsis line (OE) with the highest expression level, *agb1-2* mutant, and the restorer line of *agb1-2* mutant (Res) for follow-up experiments (Appendix A). The seeds from wild-type Arabidopsis (WT), OE, Res, and *agb1-2* lines, were evenly seeded in the MS medium, without additives or containing 9% PEG6000, 12% PEG6000, 100 mM NaCl and 150 mM NaCl, respectively. We found that the seed germination rates of WT, overexpression *TaGB1-B* Arabidopsis line (OE), mutant Arabidopsis(*agb1-2*), and the restorer line of *agb1-2* mutant (Res) on the MS medium were not significantly different (Figure 3). In MS supplemented with PEG6000, the seed germination rates of several lines were inhibited, and the germination rate of all lines was significantly inhibited on the MS medium containing NaCl (Figure 4). In general, the germination rate of OE and Res lines is higher than that of WT and *agb1-2*. The germination rate of OE is the highest overall, and *agb1-2* is the lowest (Figure 3 and Figure 4).

After that, we conducted root length experiments and evaluated the phenotypic changes of these lines. For samples cultured on the MS medium, the root length of Arabidopsis lines did not change significantly (Figure 5). However, the growth of each line was significantly limited in the MS medium supplemented with PEG6000 or NaCl, and fresh weight also decreased, and the root length and fresh weight of OE and Res lines were significantly higher than those of WT and *agb1-2*. Therefore, the *TaGB1-B* gene may affect the ability of Arabidopsis to resist drought and salt stress. 

To explore whether *TaGB1-B* is involved in plant stress resistance at the seedling stage, we conducted drought and salt treatments in the Arabidopsis seedling stage. We counted the survival rate of the four lines and measured the corresponding proline (Pro) and malondialdehyde (MDA) contents. We found that there was no difference among the lines before treatment (Figure 6a and Figure 7a). After treatment, all Arabidopsis lines were stressed. After two weeks of drought stress, *agb1-2* withered seriously. However, the effect of drought stress on OE and Res is relatively little effect. After 3 days of re-watering, OE and Res continued to grow, while *agb1-2* died as the leaves turned green. The survival rates of OE and Res were 88% and 86%, much higher than WT (68%) and *agb1-2* (45%) (Figure 6b). The MDA content of OE was significantly lower than WT, while the Pro content was higher than WT (Figure 6c,d). These results showed that overexpression of *TaGB1-B* enhanced the drought tolerance of Arabidopsis. After salt stress, WT and *agb1-2* were undergoing serious chlorosis, while OE and Res were slightly withered. The survival rate of OE (66%) and Res (66%) was higher than that of WT (38%) and *agb1-2* (22%) (Figure 7b). After stress, the content of Pro in OE and Res was significantly higher than WT and *agb1-2*, and the content of MDA was significantly lower than WT and *agb1-2* (Figure 7c,d). The above results showed that the overexpression of *TaGB1-B* enhanced the salt tolerance of Arabidopsis.

### 2.4. Overexpression of TaGB1-B Enhances the Ability of Wheat to Resist Drought and Salt Stress

In order to determine the role of *TaGB1-B* in drought and salt tolerance of wheat, transgenic lines of wheat variety KeNong 199 (KN199) overexpressing *TaGB1-B* and an empty vector gene were produced, and T_3_ transgenic homozygous lines were obtained and detected by qRT-PCR. We selected three T_3_ generation *TaGB1-B* overexpression wheat transgenic lines (TaOE3, TaOE4, TaOE5) with high expression levels, and one T_3_ generation transgenic wheat line with an empty vector (EV) for follow-up experiments (Appendix A). We evaluated the expression level of *TaGB1-B* under drought and salt stress in one-week-old overexpression wheat seedlings. The data indicated that under stress conditions, the expression levels of *TaGB1-B* genes were upregulated in WT and transgenic plants, but the expression levels were higher in the *TaGB1-B* overexpressing plants than in the control plants (Figure 8). Under drought stress, the expression of *TaGB1-B* reached its peak at 24 h, and then showed a downward trend. The expression of *TaGB1-B* in TaOE3 was the highest at 24 h, being about 6.5 times higher (Figure 8a). Under salt stress, the expression of *TaGB1-B* decreased first and then increased. The expression of *TaGB1-B* in TaOE3 was the highest at 48 h, being about 3.4 times higher (Figure 8b). 

Considering that the root system is related to the abiotic stress resistance of plants, we also examined the morphology of the roots and found that there are no obvious differences in root length and lateral root number between the KN199 and transgenic wheat with the empty vector gene (EV) under various growth conditions (Figure 9). This indicates that the empty vector gene transferred into wheat has no effect on the root growth of wheat. Under normal growth conditions, there was no significant difference in the root length between KN199 and *TaGB1-B* overexpressing wheat transgenic lines (TaOE3, TaOE4, TaOE5) (Figure 9b), but the number of lateral roots of KN199 was significantly higher than that of overexpression wheat (Figure 9c). After the treatment of drought and salt stress, the root length and lateral root number of the overexpression line were significantly higher than that of KN199, and the root length and lateral root number of TaOE3 were the highest as a whole. Taken together, these results indicated that the overexpression of *TaGB1-B* increased the root length and lateral root number of wheat under drought and salt stress, and improved the ability of wheat to resist drought and salt stress.

In order to explore whether *TaGB1-B* is involved in the stress resistance of wheat seedlings, we conducted drought and salt treatment on two-week-old wheat seedlings. We measured several physiological indexes and counted the survival rate of five lines after rehydration. Under normal conditions, no visible phenotypic difference was observed between the OE lines, EV and WT, and the phenotypes and physiological indexes of EV and WT had no significant difference under stress treatments (Figure 10 and Figure 11). Drought stress had little effect on the overexpression lines, and the growth of OE lines was better than that of WT and EV controls (Figure 10a). After drought treatment, physiological indicators had an increasing trend, but the trend is different. The MDA content of several lines decreased, and the PRO content and the activities of superoxide dismutase (SOD) and peroxidase (POD) increased, but the MDA content in OE was lower than WT, and the PRO content and SOD content were higher than WT (Figure 10c–f). In addition, the survival rate of OE after rehydration was higher than that of KN199 (Figure 10b). After salt stress treatment, both KN199 and overexpression plants showed some degree of wilting and inhibited growth, but the wilting degree of overexpression plants was significantly lower than that of KN199 (Figure 11a). The changes in several physiological indicators were basically consistent with the drought treatment (Figure 11). These results showed that the overexpression of *TaGB1-B* increased the tolerance of wheat variety KN199 to drought and salt at the seedling stage. 

## 3. Discussion

The hexaploid wheat genome (2n = 6x = 42, AABBDD) is composed of three subgenomes, A, B and D, and is extremely complex. More than 80% of the sequences in the three subgenomes are similar [36]. Gawande used the dataset from the expression database to compare the expression of *TaGB1* homologs in all 71 tissues and the development time points, and found that the level of expression for the A, B and D homologs was similar. In 71 tissues and development stages, all three homologs showed relatively similar levels of expression [37]. In this study, *TaGB1* showed a high level of homology across the A, B and D chromosome groups, and its expression levels are similar under different stress conditions (Figure 1). This suggests that *TaGB1* is a highly conserved gene across the wheat chromosome groups.

TaGB1 contains seven WD40-conserved domains and is a member of the WD40 protein family. WD40 proteins are involved in various processes, such as growth and development, metabolite biosynthesis, and immune and stress responses [13,21,38,39,40,41,42,43,44,45]. Furthermore, it has been shown in Arabidopsis [15,46,47], rice [29,48,49,50], maize [51,52,53], and rapeseed [54,55] that G-proteins are involved in abiotic stresses. In this study, the qRT-PCR analysis showed that the expression of *TaGB1-B* changed under stress and hormone treatments (Figure 2a–i). This finding is consistent with previous studies. Further analysis of *TaGB1-B* expression under drought and salt treatment showed that its expression was regulated differently in different tissues. Specifically, drought treatment upregulated the expression of *TaGB1-B* in roots and leaves, while *TaGB1-B* expression in roots, stems, and leaves showed almost no difference under salt stress, but showed an upward trend instead (Figure 2j,k). These results indicate that *TaGB1-B* can regulate the response of wheat to abiotic stress.

Seed germination rate is one of the important indicators for measuring the quality of crop seeds, which directly affects the yield and quality of crops. A high seed germination rate means more seeds can successfully germinate into seedlings, thereby increasing the crop emergence rate and growth speed, and improving crop yield and quality. In addition, the seed germination rate also affects the entire growth season and the maturity time of crops, thereby affecting the harvest and efficiency of agricultural production. Therefore, ensuring a high seed germination rate is one of the important measures to improve crop yield and quality [56,57,58,59]. The root system is the main organ for nutrient uptake in plants, capable of absorbing water, nutrients, and other essential substances. A well-structured and functional root system can enhance plant growth rate and yield, as well as increase the plant’s ability to withstand adversity. Environmental disruptions are closely related to changes in the root structure. In order to adapt to adversity, plants can regulate the growth of primary roots, lateral roots, or adventitious roots, as well as the length and distribution of root hairs [60,61]. Existing research has shown that under non-biological stress, Arabidopsis thaliana strains with transferred related genes have higher germination rates and longer roots than wild-type plants [62,63,64]. This situation has also been observed in wheat overexpressing *TaSNAC8-6A* [65], transgenic Arabidopsis overexpressing *AtNAC2* [15,66], and transgenic rice overexpressing *OsNAC6* [67]. In our experiment, the overexpression of *TaGB1-B* in Arabidopsis resulted in higher germination rates and longer primary roots under drought and salt stress compared to WT and *agb1-2* mutant lines (Figure 3, Figure 4 and Figure 5). The situation in wheat is similar to that in Arabidopsis, with overexpression leading to a longer root length and more lateral roots than the KN199 (Figure 9). Our findings suggest that *TaGB1-B* plays an important role in regulating root development and stress responses, and may be a potential target for improving plant growth and stress tolerance. 

Malondialdehyde (MDA) is one of the commonly used indicators to measure the degree of oxidative stress and reflects the degree of membrane lipid peroxidation in plants. Proline (Pro) is one of the components of plant proteins and can exist widely in a free state in the plant body, and accumulated Pro acts as an osmoregulatory substance in plant cytoplasm [44,68,69]. When plants are subjected to abiotic stress, the higher the oxidative stress response, the higher the MDA content. Conversely, for the soluble substance Pro in the cells, the higher the content, the stronger the protective effect on plants [62,64,70]. Our research results indicate that under drought and salt stress, the MDA and Pro contents of all lines increased significantly, but the MDA content of the transgenic Arabidopsis and wheat was lower than that of the WT, and the Pro content was higher than that of the WT (Figure 6 and Figure 7). This indicates that transgenic plants overexpressing *TaGB1-B* have stronger drought and salt tolerance.

Among the physiological issues that accompany abiotic stresses in plants, the excessive accumulation of ROS (particularly O^2−^ and H_2_O_2_) and high concentrations of ROS lead to the damage of proteins, lipids and nucleic acid, and eventually lead to cell damage or death [71]. A large amount of evidence shows that drought and salt stress change the number and activity of enzymes involved in scavenging oxygen free radicals. G-protein can reduce the damage of reactive oxygen species to cells by limiting the production of ROS or enhancing the detoxification mechanism, thus enhancing the adaptation to abiotic stress. Plants have evolved a suite of corresponding protection strategies (i.e., enzymatic and non-enzymatic anti-oxidative systems) to protect themselves from oxidative damage [72]. Antioxidant enzymes (AE), such as SOD, CAT and POD, are critical mediators of ROS detoxification caused by environmental stressors. In recent years, it has been confirmed that the transcription of different AE genes is regulated by G-protein subunits. For example, the transcripts of the CAT gene in rice are regulated by *RGG1*, a γ subunit of the G-protein. An overexpression of *RGG1* can enhance CAT activity and salt tolerance of plants [29]. The ectopic expression of rice *RGB1* in transgenic lines enhanced the salt tolerance of rice and the expression level of enzymes involved in antioxidant stress defense (such as superoxide dismutase (SOD) and ascorbic acid peroxidase (APX)) increased in the overexpression lines [73]. These findings suggest a relationship between the G-protein subunit and ROS homeostasis in plant cells. Our results showed that after drought treatment, the activities of superoxide dismutase (SOD) and peroxidase (POD) increased in several wheat lines, but the SOD content was higher in OE than in WT (Figure 10). After salt stress treatment, the changes in several physiological indicators were similar to those observed under drought treatment (Figure 11). These results suggest that *TaGB1-B* may play an important role in conferring drought and salt tolerance to wheat.

In summary, our study provides evidence for the involvement of the *TaGB1-B* gene in enhancing drought and salt tolerance in Arabidopsis and wheat. Our results suggest that *TaGB1-B* may be a promising candidate gene for improving drought and salt tolerance in wheat. Further research is needed to elucidate the molecular mechanisms of *TaGB1-B* gene expression regulation, and explore its potential applications in crop improvement programs.

## 4. Materials and Methods

### 4.1. Plant Materials and Growth Conditions

Wheat seeds (“KN199” and “Chinese Spring”) and wild-type Arabidopsis (Col-0) were from our laboratory (the laboratory for genetic improvement of wheat stress resistance, Northwest Agricultural and Forestry University). Both wheat and Arabidopsis were cultured at 20–25 °C, with a light cycle of 16 h of light/8 h of dark, light intensity of 15000 Lux, and humidity of about 70%.

### 4.2. Sequence Analysis and Expression of TaGB1 Homologous Gene

We used transcriptome data from the WheatEXP website (https://wheat.pw.usda.gov/WheatExp/, accessed on 24 March 2021) to study the expression pattern of *TaGB1* homologues in response to abiotic stress. The RNA-seq data comes from nine different growth stages and tissues of Chinese Spring under normal conditions, as well as the following six kinds of stress and normal control (heat stress 1 h, drought stress 1 h, heat stress 6 h, drought stress 6 h, combined drought and heat stress 1 h, combined drought and heat stress 6 h, no threat control) (Appendix A). The cDNA-coding region of *TaGB1-B* was isolated from CS. The primers used are shown in Appendix A.

### 4.3. RNA Extraction and qRT-PCR Analysis

The Chinese Spring was placed evenly in the culture dish, and when it was cultivated to the trefoil stage (about 10 days), drought, NaCl (300 mmol/L), PEG6000 (20%), cold, heat (38 °C), ABA (100 μmol), GA (100 μmol), MEJA (100 μmol) and ET (100 μmol) treatments were carried out. Leaves were taken at 0 h (CK), 0.5 h, 1 h, 3 h, 6 h, 9 h, 12 h and 24 h, and the roots, stems and leaves after drought and salt treatment at 0 h (CK), 1 h, 3 h, 6 h, 12 h and 24 h. Then, they were quickly frozen with liquid nitrogen and stored in a refrigerator at −80 °C for standby. The total RNA was extracted with the GenStar (GenStar, Beijing, China) Trizol kit, and cDNA was synthesized with the TransGen (TransGen, Beijing, China) reverse-transcription kit. The cDNA of wheat under two stress treatments was used as the template, and *β-Actin* was used as the internal reference gene. qRT-PCR was performed with the SYBR Green dye method to analyze the expression of the *TaGB1* gene. Three replicates were set for each sample. The relative expression of each gene was calculated according to the 2^−∆∆CT^ value. The primers used in this experiment are shown in Appendix A.

### 4.4. Arabidopsis Transformation and Stress Tolerance Assays

In order to obtain the overexpressed plant of Arabidopsis, we connected the sequence of *TaGB1-B* to the pCAMBIA1302 vector. The recombinant plasmid with the correct sequence was transformed into Agrobacterium line GV3101, and then transformed into the Arabidopsis Col-0 type using the inflorescence immersion method. The mutant and overexpression information of Arabidopsis are shown in Appendix A. Then, the *TaGB1-B* overexpression seeds were cultured to the T_3_ generation. The growth conditions of Arabidopsis were the same as for the wheat. We cultured the seeds of Arabidopsis WT, *TaGB1-B* overexpression, *agb1-2* mutant and complementary *agb1-2* mutant lines on the MS medium containing PEG6000 (9% and 12%), NaCl (100 mM and 150 mM), and MS with no additives, and recorded the germination rate every 12 h.

The same growth state Arabidopsis seedlings were cultured vertically on the MS medium containing 12% PEG6000 or 100 mM NaCl, and the root length was recorded after seven days of growth. In order to test the drought tolerance of Arabidopsis, we transferred Arabidopsis seedlings with the same growth state on the MS into the soil. After three weeks of culture, the seedlings were subjected to drought stress (no water supply) for two weeks, and then rehydrated for three days. The survival rate and the contents of MDA and Pro were recorded. In order to investigate the salt tolerance, after three weeks of culture, plants were irrigated with 200 mM NaCl for two weeks, and the survival rate and the contents of MDA and Pro were recorded. We performed three independent biological replications.

### 4.5. Wheat Transformation and Stress Tolerance Assays

Callus was induced from the young embryos of KeNong 199 which were pollinated for about 14 days. The constructed pWMB003-*TaGB1* and pWMB003 empty vector plasmid were bombarded into the calli by gene gun bombardment, and then differentiated and screened. Transgenic-positive and empty vector (EV) transgenic line plants in each generation were screened by PCR. The relative expression of *TaGB1-B* in transgenic plants was detected by using the methods of RNA extraction, cDNA synthesis and qRT-PCR, described above. 

In order to determine the effect of drought and salt stress on the root at the wheat seedling stage, the wheat seedlings of WT, EV and OE lines, at the age of one week, were cultured with 1/2 Hoagland nutrient solution plus 20% (M/V) polyethylene glycol (PEG6000) and 1/2 Hoagland nutrient solution with 300 mM NaCl, respectively. After one week of treatment, the morphological and physiological parameters of the plant roots were measured.

In order to analyze the drought tolerance of wheat at the seedling stage, WT, EV and OE seeds were germinated in Petri dishes and transplanted into the greenhouse for 16 h of light/8 h of darkness for two weeks. When the relative water content of the soil is about 60%, the drought stress is carried out (no watering for two weeks), and then re-watering takes place for 3 days. The phenotype and survival rate were recorded and the physiological indexes of each wheat line under drought stress were determined. In order to analyze the salt tolerance of wheat seedlings, the salt tolerance experiment was carried out on the wheat growing for three weeks. The wheat was irrigated with 300 mM NaCl for three weeks. The plant survival rate and physiological indexes of each wheat line were recorded. Three independent biological replications were carried out in the above experiments.

### 4.6. Statistical Analysis

We used the SPSS (Chicago, Illinois, USA) software for statistical analysis. The data were calculated with the mean ± standard deviation (SD) and an analysis of variance (ANOVA), and the significance level was defined as * (*p* < 0.05) and ** (*p* < 0.01).

## 5. Conclusions

In this study, we selected *TaGB1-B* for subsequent functional verification. The results showed that *TaGB1-B* improved the tolerance of Arabidopsis and wheat plants to drought and salt stress. This work provides a theoretical basis for wheat G-protein β-subunits in a further study, and new genetic resources for the cultivation of drought-tolerant and salt-tolerant wheat varieties. This work has a strong future application value, especially considering the expected crop losses related to climate change.

## Figures and Tables

**Figure 1 ijms-24-07337-f001:**
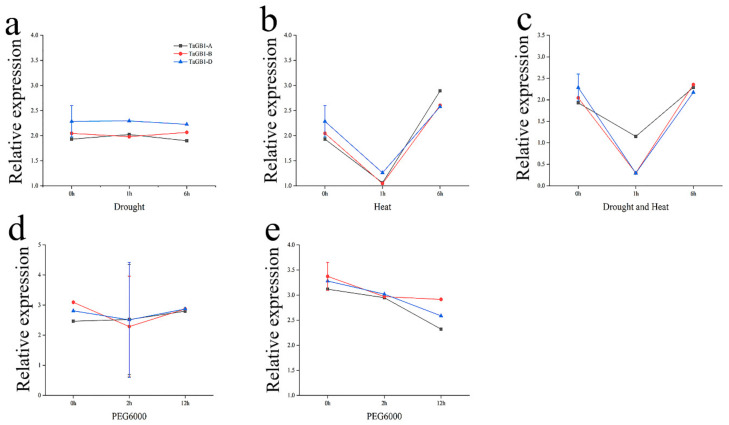
The expression of *TaGB1* homologs under different stress treatments in one-week-old wheat seedling (based on the wheat expression database). (**a**) Expression of *TaGB1* homologs under drought stress in wheat variety TAM107. (**b**) Expression of *TaGB1* homologs under heat stress in wheat variety TAM107. (**c**) Expression of *TaGB1* homologs under drought and heat stress in wheat variety TAM107. (**d**) Expression of *TaGB1* homologs under PEG6000 stress in wheat variety Gemmiza 10. (**e**) Expression of *TaGB1* homologs under PEG6000 stress in wheat variety Giza 168.

**Figure 2 ijms-24-07337-f002:**
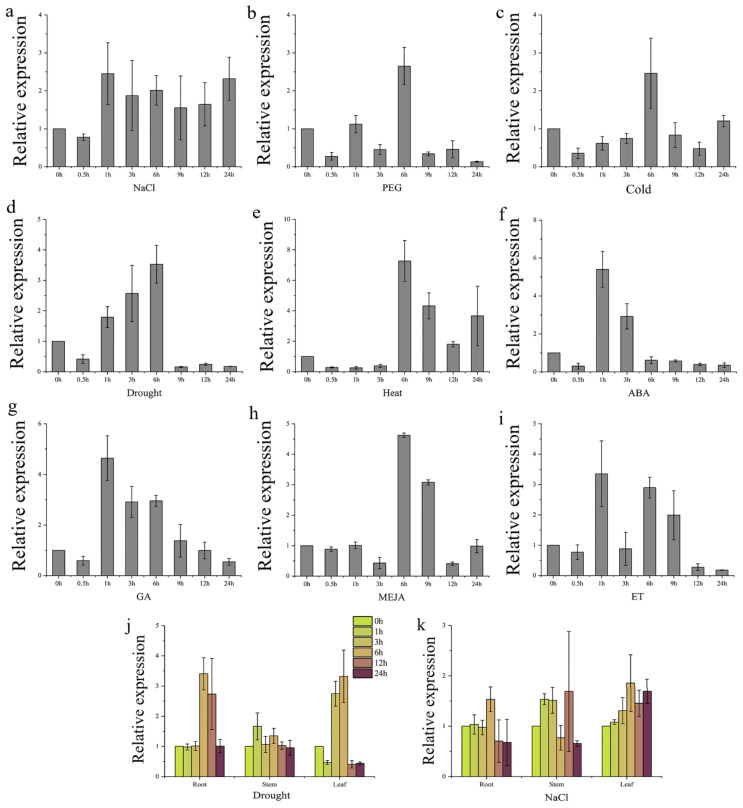
The expression of *TaGB1-B* under different stress treatments and tissue-specific expression of *TaGB1-B* under drought and salt treatments in one-week-old wheat seedling. (**a**) The expression of *TaGB1-B* under NaCl treatment; (**b**) PEG6000; (**c**) cold; (**d**) drought; (**e**) heat; (**f**) ABA; (**g**) GA; (**h**) MeJA; and (**i**) ET. (**j**) Tissue-specific expression of *TaGB1-B* under drought treatment. (**k**) Tissue-specific expression of *TaGB1-B* under salt treatment. Error bars show standard deviations (mean ± SD and n = 3).

**Figure 3 ijms-24-07337-f003:**
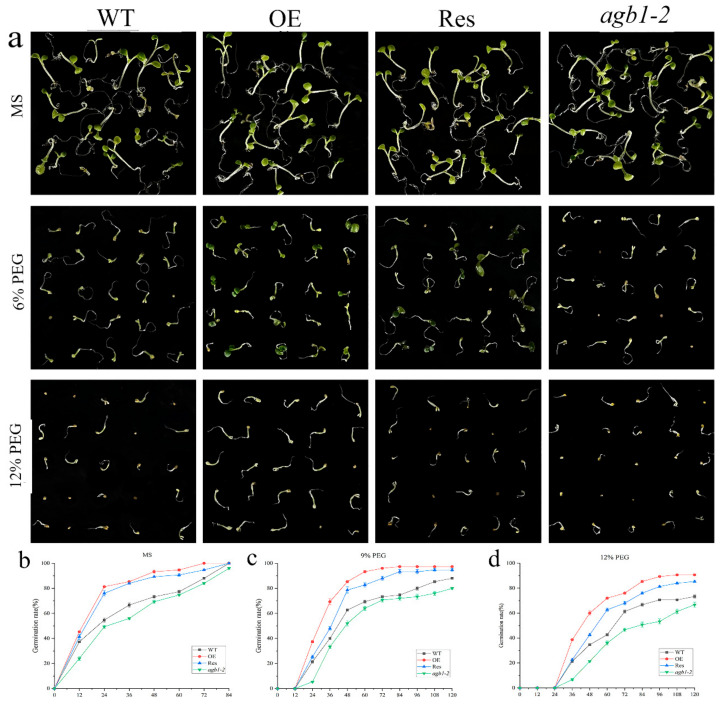
Germination assay of wild-type (WT), *TaGB1-B* overexpression Arabidopsis, mutant Arabidopsis (*agb1-2*) and complementary mutant (Res) seeds under PEG6000 treatment. (**a**) The phenotypes of WT, *TaGB1-B* overexpression Arabidopsis, mutant Arabidopsis (*agb1-2*) and complementary mutant seeds under 9% and 12% PEG6000 treatments. (**b**) The germination rates of WT, *TaGB1-B* overexpression Arabidopsis, mutant Arabidopsis (*agb1-2*) and complementary mutant seeds at different time points on the MS medium. (**c**) The germination rates under 9% PEG6000 treatment. (**d**) The germination rates under 12% PEG6000 treatment.

**Figure 4 ijms-24-07337-f004:**
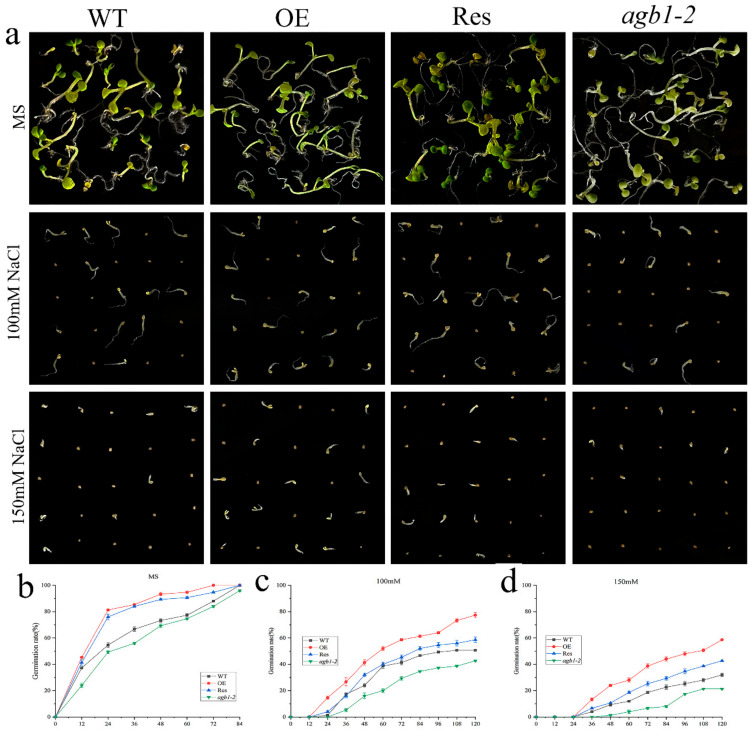
Germination assay of wild-type (WT), *TaGB1-B* overexpression Arabidopsis, Arabidopsis mutants (*agb1-2*) and complementary mutant (Res) seeds under NaCl treatment. (**a**) The phenotypes of WT, *TaGB1-B* overexpression of Arabidopsis, mutant Arabidopsis (*agb1-2*) and complementary mutant seeds under 100 mM and 150 mM NaCl treatments. (**b**) The germination rates of WT, *TaGB1-B* overexpression of Arabidopsis, mutant Arabidopsis (*agb1-2*) and complementary mutant seeds at different time points on the MS medium. (**c**) The germination rates under 100 mM NaCl treatment. (**d**) The germination rates under 150 mM NaCl treatment.

**Figure 5 ijms-24-07337-f005:**
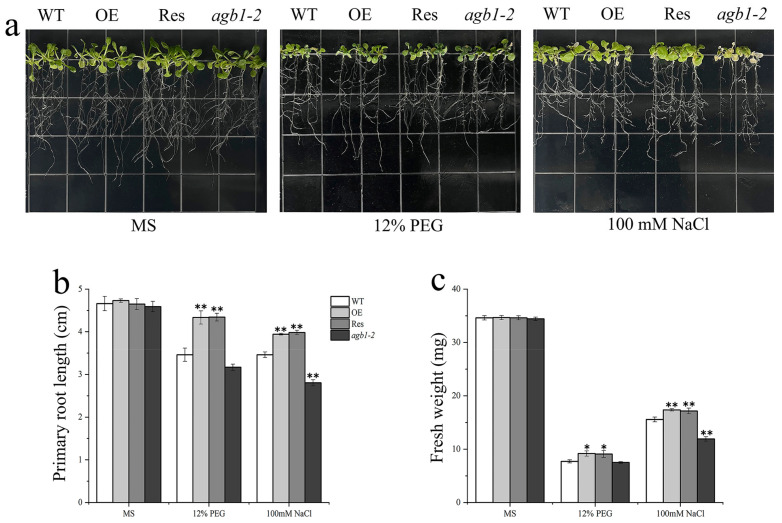
The overexpression of *TaGB1-B* enhanced the tolerance to drought and salt stresses in Arabidopsis. (**a**) Morphological differences between wild-type (WT), *TaGB1-B* overexpression Arabidopsis, mutant Arabidopsis (*agb1-2*) and complementary mutant seeds at seedling stages under PEG6000 treatment and NaCl. (**b**) Statistical analysis of root length. (**c**) Fresh weight. The error column represents the standard deviation (mean ± SD and n = 3), * and ** above each column indicate a significant difference compared with WT plants (* *p* < 0.05; ** *p* < 0.01).

**Figure 6 ijms-24-07337-f006:**
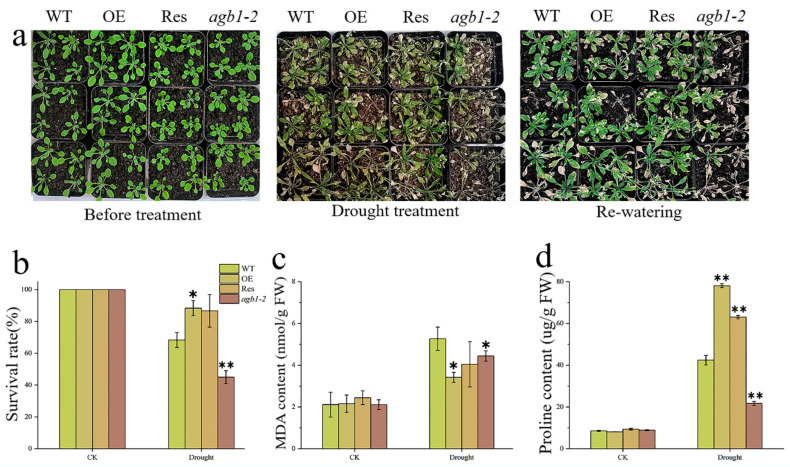
The functional characteristics of drought tolerance of transgenic Arabidopsis with *TaGB1-B* gene under drought stress. (**a**) Phenotypic analysis of each plant line before treatment, 14 days after drought treatment, and 3 days after rehydration. (**b**) Survival rate of each line after drought stress. (**c**) The content of malondialdehyde (MDA) in Arabidopsis treated by drought. (**d**) Proline content. The data are expressed as the mean ± SDs (n = 3) of three experiments. * and ** above each column indicate a significant difference compared with WT plants (* *p* < 0.05; ** *p* < 0.01).

**Figure 7 ijms-24-07337-f007:**
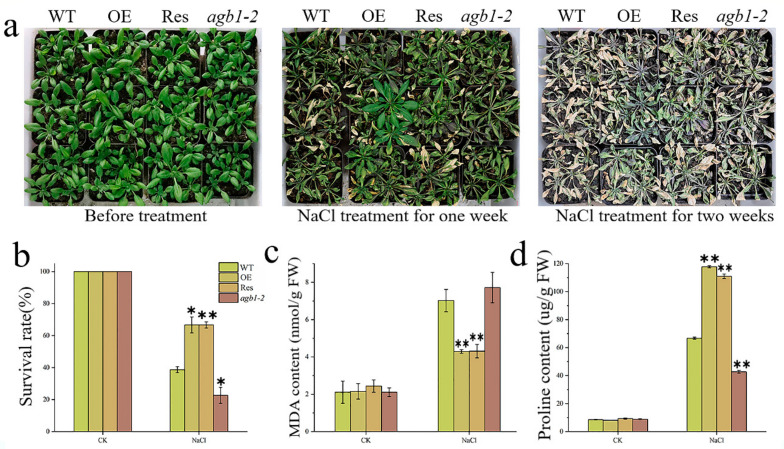
The functional characteristics of drought tolerance of transgenic Arabidopsis with *TaGB1-B* gene under NaCl stress. (**a**) Phenotypic analysis of each plant line before treatment, one week after NaCl treatment and two weeks after NaCl. (**b**) Survival rate of each line after NaCl stress. (**c**) The content of malondialdehyde (MDA) in Arabidopsis treated by NaCl. (**d**) Proline content. The data are expressed as the mean ± SDs (n = 3) of three experiments. * and ** above each column indicate a significant difference compared with WT plants (* *p* < 0.05; ** *p* < 0.01).

**Figure 8 ijms-24-07337-f008:**
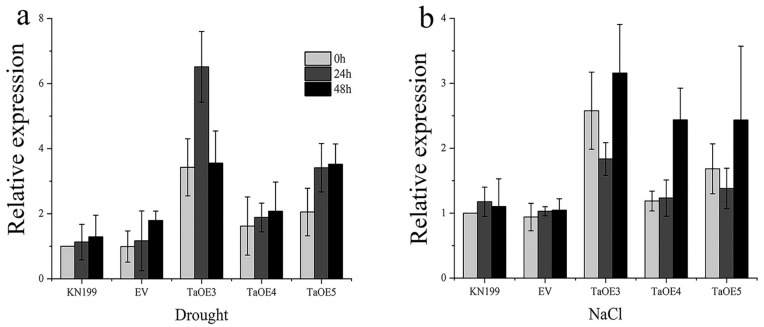
Transcription spectrum analysis of *TaGB1-B* transgenic wheat under drought and salt stress. (**a**) The expression level of *TaGB1-B* in each line under drought treatment. (**b**) The expression level of *TaGB1-B* of each line under NaCl treatment. Error bars show standard deviations (mean ± SD and n = 3).

**Figure 9 ijms-24-07337-f009:**
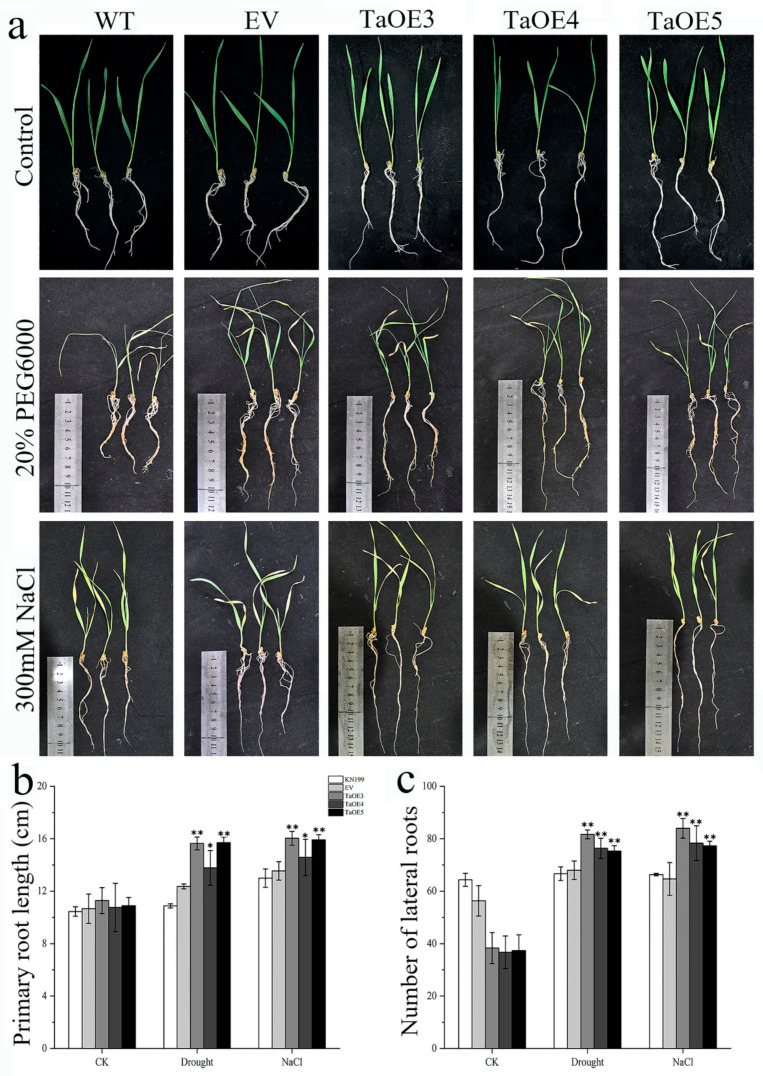
*TaGB1-B* enhanced the drought and salt tolerance of wheat. (**a**) Drought and salt stress tolerance responses of *TaGB1-B* overexpression transgenic KN199 and the empty vector transgenic plants. (**b**) Primary root length. (**c**) Number of lateral roots. Data are presented as the mean ± standard deviation calculated from triplicates. * and ** above each column indicate a significant difference compared with WT plants (* *p* < 0.05; ** *p* < 0.01).

**Figure 10 ijms-24-07337-f010:**
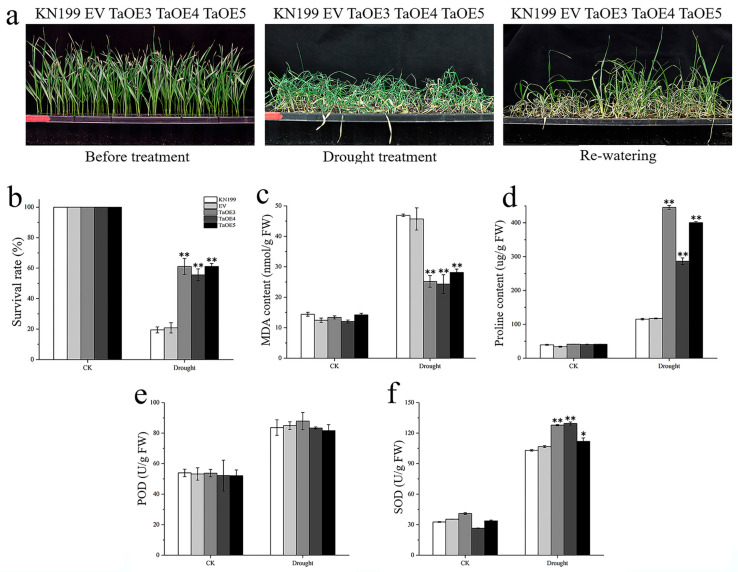
The functional characteristics of *TaGB1-B* overexpression wheat plants under drought stress. (**a**) Phenotypic analysis of *TaGB1-B* overexpression plants under normal treatment, drought stress treatment for 14 days and rehydration for 3 days. (**b**) The survival rate of each line after drought stress. (**c**–**f**). Physiological index of wheat plant. (**c**) MDA content. (**d**) Proline content. (**e**) POD. (**f**) SOD. The error column represents the standard deviation (mean ± SD and n = 3), and * and ** above each column indicate a significant difference compared with WT plants (* *p* < 0.05; ** *p* < 0.01).

**Figure 11 ijms-24-07337-f011:**
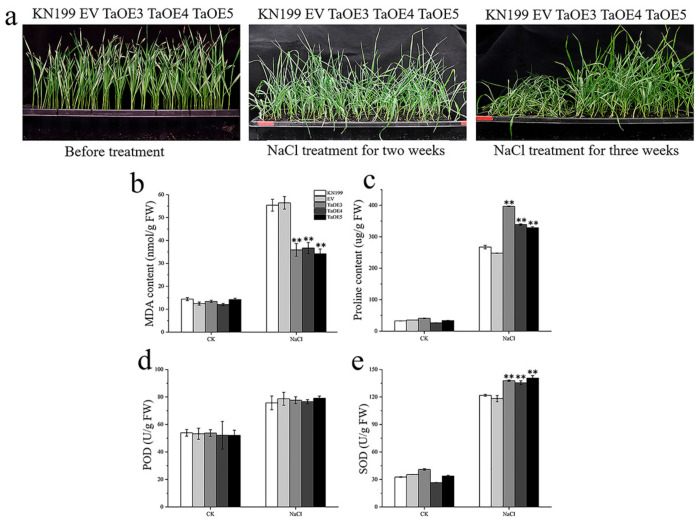
The functional characteristics of *TaGB1-B* overexpression wheat plants under NaCl stress. (**a**) Phenotypic analysis of *TaGB1-B* overexpression plants under normal treatment, NaCl stress treatment for two weeks and NaCl stress treatment for three weeks. (**b**–**e**) Physiological index of wheat plant. (**b**) MDA content. (**c**) Proline content. (**d**) POD. (**e**) SOD. The error column represents the standard deviation (mean ± SD and n = 3), and and ** above each column indicate a significant difference compared with WT plants (** *p* < 0.01).

## Data Availability

Not applicable.

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
