# Peer review of "G-Protein β-Subunit Gene TaGB1-B Enhances Drought and Salt Resistance in Wheat"

_ijms, 2023, doi:10.3390/ijms24087337_

Round 1
Reviewer 1 Report
The work is interesting and provides a theoretical basis for wheat G protein β subunits in further study and new genetic resources for the cultivation of drought-tolerant and salt-tolerant wheat cultivars. Various and adequate analyzes have been applied, from which interesting data have been obtained.
Some remarks:
Line 91 - "Wang compared the homolog..." must be Wang and coworkers compared the homology...
Line 93 - "Nilesh used the data set..." must be Gawande et al. used the data set from the expression database to compare the expression of TaGB1 homologs in all 71 tissues and development time points [36].
Line 348 and other places - "qRT-PCR" better to be RT-qPCR.
The word "indicate" is repeated four times just in the abstract.
The first few sentences in the Results section are not the authors' results and must be in the Discussion section.
Figure 8 b - I'm afraid I have to disagree with the star's position. Please check again!
Reviewer 2 Report
Dear editor and colleagues,
I have read with interest the manuscript “G Protein β Subunit TaGB1-B Enhances Drought and Salt Resistance in Wheat” submitted in IJMS.
It is a study that focuses on the role of G Protein for abiotic stress in soft wheat.
The authors have used appropriate techniques (produced transgenic Arabidopsis and wheat plants and followed the relative transcription and metabolite induction after abiotic stresses).
The results obtained constitute novel data. Also, the conclusions drawn are justified by the results. Hence the manuscript has merit for publication.
In terms of experimental designing, I have little to comment.
Still there are some things that the authors could focus on a revised version.
· The use of English language is not clear across the text, and it makes it difficult to follow at several instances. Proofreading is strongly advised.
· Regarding the statistics, several figures do not show the anova groups/significance across bars.
· Some parts in the results should be relocated to the discussion part (for instance L91-93 etc)
· The authors should double check gene/protein nomenclature across text. Protein (capital/no italics), gene (capital, italics), mutant (no capital etc)
· The discussion part is rather weak and restricted.
Also, as a question to the authors. Usually regarding gene functional characterization besides over expression, a knocked out is also attempted. Why weren’t TaGB1-B lines produced?
Based on the above I recommend a major revision
Reviewer 3 Report
Dear Author
The manuscript address an important issue such as drought and salinity. However, Writing style is not suitable for international publication. I highly recommend to rewrite the manuscript and submit it again. For example:
line 28, Wheat (Triticum aestivum L.) was affected by various abiotic stresses during its 28 growth and development, such as drought, salinity, heat, cold, heavy metals, and ozone, 29 etc. [1]. was affected? or is affected? Do you mean during its evolution?
Line 91, who is Wang? A previously published work? one of the co-authors?
most importantly you wrote the manuscript in present tens while it should be past tens as passive sentences.
Best wishes
Round 2
Reviewer 2 Report
the authors have adressed most of my comments i believe the ms can be accepted